# Deployment of Deep Learning Model in Real World Clinical Setting: A Case Study in Obstetric Ultrasound

Anonymous Full Paper
Submission 44

## Abstract

Despite the rapid development of AI models in medical image analysis, their validation in real world clinical settings remains limited. Models are often developed without continuous feedback from clinicians, which can lead to a lack of alignment with the actual needs. To address this, we introduce a generic framework designed for deploying and testing image-based AI models early in such settings. Using this framework, we deployed a trained model for fetal ultrasound standard plane detection and evaluated it in real-time sessions with both novice and expert users. Feedback from these sessions revealed that while the model offers potential benefits to medical practitioners, the need for navigational guidance was identified as a key area for improvement. These findings underscore the importance of early testing of AI models in real-world settings, leading to insights that can guide the refinement of the model and system based on actual user feedback.

## 1 Introduction

The clinical community is eagerly anticipating the validation of AI in real-world clinical settings [1]. This is distinct from retrospective validation using previously recorded videos [2]. For many imaging modalities, dynamic decision-making is required not only for image recognition (i.e. "what am I looking at?") but also for image acquisition (i.e. "where to look at in the first place?"). Furthermore, AI systems are anticipated to face more practical challenges in real-world clinical settings [3–5]. Lessons learned from case studies studying the deployment of AI tools for clinical applications highlight that well-performing AI models may fail for unexpected reasons in the real world. For instance, Beede et al. [6] studied the deployment of a diabetic retinopathy detection system with >90% sensitivity and specificity in the lab but faced severe ungradability issues in a real-world setting. Their system refused to grade 21% of the images citing quality issues, although the images were acceptable to human readers, introducing unnecessary delay in a busy clinic. This underscores that the actual utility and value of an AI model remain unclear until it is tested under real-world conditions.

Real-world clinical deployment is challenging. First, the success of an AI tool in the clinic strongly depends on how well it integrates with the clinical workflow [7]. However, researchers are often not allowed to deploy developmental work directly into a medical device in the clinic for security reasons, effectively creating an upper bound on how well-integrated the deployment can be. Second, existing deployment tools focus on making the inferencing pipeline efficient and streamlined, while research code is often messy. These factors lead to overhead, discouraging AI researchers from deploying their models early in their development process. However, we advocate for testing deep learning models in the clinical setting as early as possible. If things should fail, they should fail early.

In this paper, we introduce a framework for the deployment of dynamical image-based AI systems from research in a clinical setting. As a case study, we illustrate our framework in the setting of fetal ultrasound standard plane detection, where despite active development of AI methods [8–11], actual deployment in the real world is rarely seen. We aim to be as integrated into the clinical workflow as possible, expecting only the HDMI output from the medical device. We discuss the constraints and present our design solution, aiming to lower the entry barrier of deploying and testing machine learning models directly from research, without the burden of making it efficient or optimized. We aim to speed up the development cycle, gain initial user feedback, refine development goals, and iterate. We describe how, using our designed solution, we deployed an explainable AI model for fetal ultrasound standard plane detection, and invited clinical practitioners to use the system as they scan their patients. Finally, we also report our findings from interviews with clinical practitioners using our deployed system. This study highlights a significant step towards bridging the gap between research and practice in the field of medical image analysis.

## 2 Method

### 2.1 Design challenges & requirements

Designing a generic framework for deploying image-based AI systems in a clinical setting presents several challenges and requirements. These include:

**Device Output** The system should not expect any

output from the medical device other than a video feed via an HDMI cable. This is because any other form of output may not generalize across different medical devices.

**Prediction Latency** The system should aim to generate predictions at minimal latency. This is different from processing retrospective videos, which favors processing large amount of data simultaneously by batch inferencing. In a live supporting system, video frames from the past become irrelevant as time progresses, and therefore the system should focus on responding to new video frames as quickly as possible to provide real-time feedback.

**Local Processing** The system should be able to run the entire processing pipeline locally, since data security is crucial for many clinical applications. Furthermore, this approach incurs a lower learning overhead for researchers than a more complicated workflow, such as a remote-server-edge-client architecture.

**Wireless Display** The system should have a mechanism for showing the live results on a wirelessly-connected display device. Wired connections are not always possible in the room setting of a clinic.

**Video Recording** The system should optionally support video recording in parallel to the prediction process. This means that while the AI model is making predictions in real-time, the system should simultaneously be able to record the video feed, which is helpful for further development of the prototype model.

**Physical Setup** The physical setup should be as small and stealthy as possible, so that it does not introduce any obstructions in a busy clinic.

**Software Compatibility** On the software level, the framework should be able to accommodate research code, which is typically chaotic by nature. Ease-of-use should be prioritized over computational efficiency.

## 2.2 Design solution

Our design solution, as illustrated in Figure 1, is a robust and flexible framework that leverages a variety of technologies to capture and process real-time video streams from medical devices. Code is available at http://ANON-REPO-URL/

**HDMI-to-USB Converter Box** We use this device to capture the real-time video stream from the medical device. The converter box feeds the video stream to a small computation server and appears as a USB webcam device on the server. This setup allows us to use common software packages such as OpenCV to capture and process the video stream.

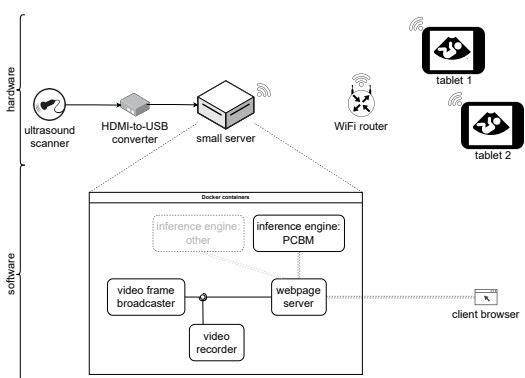

**Figure 1.** System architecture. Live video is streamed from the scanner to the server, where all processing steps are executed within respective containers. Results are subsequently rendered on a webpage, which is accessible wirelessly from tablets.

**Computation Server** The server is responsible for all computation needs of the deployment. This server can be CPU-only to satisfy economical considerations or security restrictions of the clinical authority, or equipped with a GPU to meet the computational needs of the researchers.

**Docker Containers** Within the server, we achieve our design requirements with the use of Docker containers [12]. These containers perform various tasks:

- **Video Frame Broadcaster** This container grabs video frames from the converter box via OpenCV and broadcasts them through a websocket.

- **Recorder** This container listens to the websocket and records the video, saving it into an mp4 file for retrospective AI development activity.

- **Webpage Server** This container also listens to the websocket, coordinates prediction on the live video stream, and hosts a webpage for displaying results. It acts as a prediction task manager/router, sending latest video frames to the inference engine for model predictions, and displaying results to the clinician (see Figure 2) upon receiving response from the inference engine.

- **Inference Engine** This container encapsulates all the code needed to runs the model prediction.

**Docker Compose** Manages the lifecycle (start, restart, etc.) of the containers.

**Wireless Router** Connects display devices and the server.

**Tablets** Display inference results accessible via a webpage hosted by the server.

Our design solution is underpinned by a number of key principles and considerations, which we will discuss next:

### 2.2.1 Containerization & System Stability

One of the primary benefits of our design is the use of containerization. This approach ensures that a failure at the component level, such as a runtime error in the AI inference engine, does not shut down the entire system. This means that even if one part of the system encounters an issue, other components, like video recording, continue to function normally. Furthermore, the containerized environment allows for the execution of research code in an isolated setting, making the system much more tolerant to the inherent messiness of research code. The use of Docker Compose as an orchestration tool allows the system to auto-restart failed inference engines.

### 2.2.2 Environment Isolation & Model Deployment

Containerization also inherently provides the benefit of environment isolation. This means models developed with different dependencies can be deployed on the same machine without conflicting with each other. It is not necessary for the model to be exported in a deploy-specialized format (e.g., ONNX, TorchScript), since the original research code can be executed within the isolated environment. This flexibility simplifies the deployment process and accelerates the transition from research to clinic.

### 2.2.3 Advanced Inferencing Pipeline & Workflow Management

For advanced inferencing pipelines that involve predictions with multiple models, more inference engine containers can be added. The pipeline can be manually programmed into the webpage server application. We chose not to orchestrate such workflows with existing workflow management software (e.g., Apache Airflow), which, while optimized for production environments, introduces unnecessary overhead for researchers wanting to test their prototype models in the clinic.

### 2.2.4 Distributed Execution & Performance Optimization

For advanced use cases, containerization also allows execution of components among distributed computational units. For example, latency-insensitive, computationally heavy workloads can be executed on a remote GPU server. For slow, compute-intensive models, it is possible to modify the web server application to only perform inference when the ultrasound operator has frozen the screen. This approach optimizes system performance and ensures efficient use of computational resources.

### 2.2.5 User Feedback & Result Display

By displaying the result via a simple web application, we can easily stream results to multiple clients simultaneously. This allows researchers to collect user feedback from multiple target users, such as clinical operators and patients, providing insights for system improvement.

## 3 Experiment

Using our framework, we deploy an AI model for fetal ultrasound standard plane detection. We first examine the additional latency introduced by this setup compared to running the model directly without containerization (see subsection 3.2). Then, we conduct a pilot study with clinicians using our system in a real-world clinical setting (see subsection 3.3). This helps in guiding both downstream technical development and future full-blown randomized control trails.

### 3.1 Fetal Ultrasound Standard Plane Detection

Standard obstetric trimester scans involve capturing ultrasound images of the fetal head, stomach, and femur [13]. The accuracy of this task is crucial as it impacts the downstream task of fetal weight estimation [14], which directly influences the accurate monitoring of fetal growth.

We chose to approach the standard plane detection problem with PCBM, a hierarchical variant of the concept bottleneck model [15] developed by Lin et al. [16] for fetal ultrasound scan quality assessment. It approaches the problem by emulating the step-by-step decision-making process of experts, starting with visual concepts from image segmentation and then applying property concepts directly tied to the task.

Compared to standard black-box approaches, this method is explainable, providing transparency in its decision-making process, which arguably allows for a better understanding and trust in the model's predictions. Instead of predicting whether or not an image is of a standard plane, PCBM offers additional explanation to what anatomical landmarks are present or missing (see Figure 2).

To determine the validity of these claims, we decided to deploy a trained PCBM model in the clinic. We aim to determine whether the model's explainability provides any additional value as a computer-aided detection (CAD) tool. More importantly, we want to establish whether such a tool fits well in the

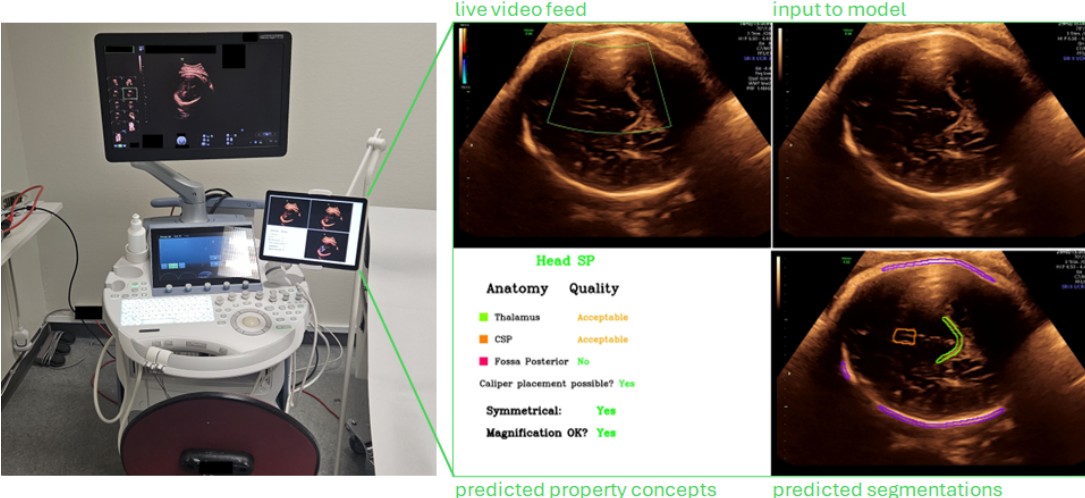

**Figure 2.** (Left) Setup at the clinic. The tablet was placed next to the scanner, while the remaining equipment was placed on a table. (Right) Screenshot of displayed prediction. A video recording is provided in the supplementary material.

clinical workflow, bringing benefits that are actually appreciated by medical workers.

## 3.2 Equipment

Following our framework as illustrated in Figure 1, we selected Magewell USB Capture Plus HDMI box as our **HDMI-to-USB converter**, TP-Link 802.11ac router (model number: TL-WR902AC) as our **WiFi router**, and two Microsoft Surface tablets as our **display devices**. We first measured the duration required to process one video frame, using three different computational devices as the **server**: a workstation (i7-7800X, Quadro P5000, TITAN V, 128GB RAM), a laptop (i7-10750H, RTX 2070 Super, 16GB RAM), and a mini PC (Intel NUC12WSKi7, i7-1260P, 32GB RAM). We also determined the extra latency introduced by our framework as compared to running the native research code directly (see Table 1). This was to understand the trade-off between the deployment ease offered by containerization and the potential increase in latency.

For our deployment, we selected the mini PC as our server to remain compliant with safety standards [17]. This was a requirement set by our deployment hospital to ensure the safety of both the patients and the medical practitioners. A photo of the setup in the clinic is shown in Figure 2.

## 3.3 Clinical sessions

We deployed the system at ANON HOSPITAL and recruited two volunteer patients in their mid-third trimester with ANON IRB's approval. We invited six novice participants (P1-P6), all senior undergraduate students enrolled in a medical program, to use our system while scanning the patients. P1 & P2 were given all explanations as predicted by PCBM. P3 & P4 were only told whether a current image is a standard plane, close to a standard plane, or at a completely unknown plane. P5 & P6 were control users without any guidance from our system. We observed the participants during the scan and interviewed them afterward to gather their feedback about our system. In separate sessions, we also invited an obstetrician (P7) and an experienced sonographer (P8) to use our tool, allowing us to gain valuable insights from a professional perspective about our system. These sessions provide an early evaluation of PCBM's performance in real clinical settings before committing to a large-scale randomized control trial study.

## 4 Results

### 4.0.1 Level of Integration into Clinical Workflow

Almost all participants expressed a desire for the prediction results to be displayed directly on the ultrasound machine. However, most participants were able to accept the current setup as a viable solution for testing purposes without being disruptive to their workflow. Meanwhile, the novice participants (P1-P4) specifically requested a higher frame rate. They expressed that a higher frame rate would allow them to move the ultrasound probe faster without the system lagging behind.

### 4.0.2 Usefulness of the Additional Explanation Provided by PCBM

P1 found the explanation on whether a specific anatomical landmark is visible helpful, while P2 took a neutral stance. Without the explanation, P3

**Table 1.** Time taken (in seconds) to process one video frame across different computational machines by running native code vs. using our framework.

| machine | Workstation | | | Laptop | | Mini PC |
|---|---|---|---|---|---|---|
| | CPU | P5000 | TITAN_V | CPU | RTX_2070S | CPU |
| native | $1.00 \pm 0.04$ | $0.25 \pm 0.04$ | $0.34 \pm 0.04$ | $2.69 \pm 0.56$ | $0.39 \pm 0.20$ | $1.19 \pm 0.09$ |
| framework | $1.06 \pm 0.04$ | $0.31 \pm 0.04$ | $0.40 \pm 0.05$ | $2.75 \pm 0.56$ | $0.45 \pm 0.21$ | $1.23 \pm 0.10$ |
| **difference** | $0.06 \pm 0.02$ | $0.06 \pm 0.02$ | $0.05 \pm 0.01$ | $0.06 \pm 0.02$ | $0.06 \pm 0.02$ | $0.03 \pm 0.01$ |

and P5 found it challenging to identify what was missing from an image before it could be considered a standard plane image.

### 4.0.3 Usefulness of the Tool in Helping a Novice User to Take High Quality Fetal Ultrasound Standard Plane Images

P1-P4 commented that the tool has helped indicating whether they are looking at a standard plane, while P5 wished for similar guidance. However, P1-P4 had difficulty in identifying which plane they were currently looking at. They adapted the strategy of blindly scanning around until the tool indicated that they were at one of the standard planes.

### 4.0.4 Additional Findings

P1, P3 & P4 expressed their wish for more navigational support. Following their strategy, the tool showed signs that they were near a standard plane every now and then, but without navigational guidance, they did not know where they should move the probe to get closer to the standard plane. They acknowledged that a higher frame rate might be helpful, but ultimately it would be ideal if the tool could tell them the direction they should move the probe if they wanted to reach a certain standard plane. This was especially the case for the femur, which had to be taken from a challenging sagittal view.

On the other hand, our interview with P7 suggested that users who are already familiar with the task may have a different use case for our tool. Instead of relying on the tool for navigational guidance, the expert used the tool for confirmation of thoughts. During the session, P7 took multiple screenshots whenever an image appeared like a standard plane image. After the session, P7 ran through the screenshots while looking at the model predictions, checking through the explanations from PCBM, and picked the best images for reporting. Meanwhile, P8 tended to rely on self-judgement rather than relying on feedback from PCBM. However, P8 commented that our predictions are generally accurate, and acknowledged that the system could be valuable for inexperienced users.

This feedback was instrumental for us in understanding the different applicability of an AI tool in a real-world setting with different types of clinical users.

## 5 Discussion & Conclusion

We have introduced a generic framework designed to deploy image-based AI models in real-world clinical settings, which focuses on research code compatibility and clinical workflow integration. Using this framework, we have successfully deployed a model for fetal ultrasound standard plane detection in a clinical environment, and evaluated its performance in real-time sessions with both novice and expert users. The feedback gathered from these sessions has provided valuable insights into the model's performance, its integration into the clinical workflow, and its potential benefits to medical practitioners.

Our findings from the interviews show that the deployed PCBM model works well as a feedback tool. However, if the intended purpose is to guide a novice user in taking better standard plane images, the tool would be an even better fit for the clinical workflow if it could provide navigational guidance. Zooming out to a bigger picture, this also emphasizes that in ultrasound, image acquisition is the major part of the challenge, which calls for different solutions than what the medical image analysis community typically focuses on [18, 19]. These findings underscore the importance of a framework that supports early deployment and testing of research models in real-world settings: Early deployment serves the crucial purpose of guiding the refinement and development of the continued technical research towards solving actually relevant clinical problems.

Leveraging our experience in this deployment, we hope to demonstrate the importance of early deployment of AI models. Early deployment leads to insights that are otherwise undiscovered, while the developmental works proceed in an undesired direction. This approach allows for the refinement of the model and system based on real-world feedback, ultimately leading to a tool that is more effective and beneficial in a clinical setting.

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
