# OpenReview forum: "Deployment of Deep Learning Model in Real World Clinical Setting: A Case Study in Obstetric Ultrasound"
_NLDL.org/2025/Conference — Submitted to NLDL 2025_

### Official Review · Reviewer_y175 · 2024-09-26
**Unsystematic collection of results, possibly out of scope, unclear definition of what is meant by a framework, no references to approvals**

**Confidence:** 4

**Summary:**

They present a framework for early testing of visual AI-tools in a clinical context and report on an experiment that applies the framework in a fetal ultrasound setting. They measured the responsiveness of the system in terms of delays and also collected (unstructured) feedback from 6 students of medicine, one obstetrician (P7) and one sonographer. While the problem is important for the utilization of deep learning models, the paper gives no contribution to deep learning as such, and does not even describe the methodology behind the system that is tested.
They describe the technical setup they have used, but it is not completely clear to me what constitutes a "framework" in the present context. They do e.g. make a point of the use wireless connections, but it is not clear to me if this should be regarded as a framework defining decision or not.
They have collected some data on the speed of the implementation compared to an off-line version, but apart from this the results they report on are "soft" in the way of summaries of the testers experiences and behaviour. No standardized questionnaires or interview guides are given. The conclusions are related to the given AI tool, rather than the framework.

**Strengths:**

The paper addresses the important issue of implementing AI tools in a real life setting, with a focus on the integration of an AI tool in a clinical work flow. They give a clear exposition of the design challenges and their chosen solution.

**Weaknesses:**

The paper may be out of scope for a deep learning conference, since the deep learning issues as such are not addressed, and the properties of the tool in question are not presented. It might fit better in health services research or operational research in health services setting.
Although the design issues and the choices made in the given solution are presented clearly, it is not obvious which design features play the role of "framework".
They give quantitative results for the delays that the AI tool has in evaluating images (which may seem too high for practical use) and compare these to an off-line version, but the paper is weak in the way of robust conclusions and findings. They quote a few pieces of feedback from the users, but these seem to be collected in an unsystematic way. A predefined questionnaire would be better, and preferably one that has been validated. The use of more "soft" feedback should at least rely on interview guides and should be analysed with methods from qualitative research. Overall, the findings appear arbitrary. Also, the findings seem to focus on the system at hand and not the framework, which is promoted as the paper's contribution.
To get it published they would also need to produce ethical approvals, but I guess these were maybe skipped in order to make the study anonymous to the reviewers.

**Final Rebuttal Confidence:**

4

**Final Rebuttal Justification:**

I am sceptical about the scientific contribution of this paper. It's unclear which parts of the technical solution constitutes the framework (e.g. the HDMI-to-USB Converter Box?) to which they claim to generalize. Also the collection of feedback and analysis thereof seems arbitrary and anecdotal. While this general issue of implementing and testing medical models is important, I think they need more than one application, a more formal definition of the framework and/or more systematic collection and analysis of feedback in order to draw generalizable conclusions. I vote for rejection although it might be suitable for a poster if they are able to describe the collection of feedback and analysis better.

**Justification:**

The distinction between specific arbitrary technical solutions and the framework is not clear. The feedback from the users was not collected in a systematic or reproducible way. The findings seem to be related to the given tool, rather then the framework, which is framed as the paper's contribution. There is no deep learning as such, which possibly places the work outside the conference's scope. They provide no ethical approvals, which are definitely needed, but this may be due to the anonymization of the manuscript.

---

### Official Review · Reviewer_L2uz · 2024-10-02
**Promising research with room for additional detail**

**Confidence:** 3

**Summary:**

The paper presents a framework for deploying a generic deep learning model, providing a case study in clinical obstetric ultrasound settings. It focuses on real-time video processing for enhancing ultrasound imaging through artificial intelligence. The authors discuss the technical infrastructure, including containerization for stability and scalability, and highlight the importance of user feedback from both novice and expert users during the deployment phase.

**Strengths:**

The paper sets out a clear objective: deploying a deep learning model in a real-world clinical setting, which is highly relevant for AI applications in healthcare.

The focus on testing the model in a real-world scenario (obstetric ultrasound) makes the work significantly relevant, especially since many AI models face challenges transitioning from lab settings to clinical environments.

Authors use an explainable AI model, addressing a key challenge in medical AI — the need for transparency in decision-making.

By gathering feedback from both novice and expert users in a clinical setting, the study emphasizes the importance of human factors in AI model deployment.

**Weaknesses:**

The paper does not seem to explicitly discuss the generalizability of the framework in detail. Although it describes the framework’s application to obstetric ultrasound, it lacks a thorough analysis of how the framework could be adapted or validated in other clinical scenarios or with different imaging modalities. For instance, how would it perform in other areas of medical imaging, such as cardiology or neurology?

The paper does not contain detailed information on the development and training of the deep learning model itself. It jumps into deployment without providing much context on how the model was trained and validated.

The dataset is limited. A larger sample size would strengthen the paper’s findings and conclusions.

The paper mainly discusses user feedback from a small group of users, without incorporating objective performance metrics of the AI model during deployment in the real-world scenario.

**Questions**

- How do you ensure the framework is generalizable to other medical imaging modalities or devices beyond fetal ultrasound?
- How well does the framework scale in terms of integrating multiple models or more complex inferencing pipelines in real-time scenarios?
- How does the latency introduced by the framework impact clinical workflow in practice? Are there any threshold levels of latency beyond which the system becomes unusable?
- You identified the need for navigational guidance in future improvements. Have you explored any strategies or approaches for providing real-time navigation to users during ultrasound scans?
- Can you provide more details on how the deep learning model was trained and validated before deployment? How did the model perform in a controlled environment? What about its performance in clinical practices?

**Extra comments**

- The labeling of subsections in Section 4 is wrong. Consider revising the formatting.
- For readability purposes, I recommend including a period “.” after each item listed in Section 2.
- The main figure illustrating the system architecture is difficult to interpret at standard size; I had to zoom in by 300% to read it properly.
- There are numerous acronyms throughout the paper (e.g., AI, HDMI, PCBM) that are not defined. Although some of them may be commonly known, there are readers who may not be familiar with them.
- Table 1 has formatting issues that make it difficult to read. I recommend adjustments in layout or style to improve its clarity. Additionally, it would be helpful to clarify what the values in the table represent (average and standard deviation?)
- In general, readability and writing style could be improved.

**Final Rebuttal Confidence:**

4

**Final Rebuttal Justification:**

The paper presents an interesting perspective to consider when implementing DL models in real-world medical applications. However, the generalizability they claimed in the proposed framework is unclear. The technical contribution of their solution seems to rely on a HDMI-to-USB converter box, which means that medical devices require to be adapted to such video format.

Additionally, the results presented are based on test surveys from user experiences collected in an arbitrary manner. There is no evaluation from a DL perspective, where the performance variation between controlled and real-world environments is analyzed. In that case, the manuscript could benefit from the user experience analysis.

**Justification:**

The paper effectively outlines a clear objective of deploying a deep learning model in a real-world clinical setting, emphasizing the significance of its application in obstetric ultrasound and the importance of transparency through an explainable AI model while considering user feedback from both novice and expert clinicians.

However, the paper lacks a detailed discussion on the generalizability of the framework, provides insufficient information on the development and training of the deep learning model, has a limited dataset that weakens its findings, and primarily relies on feedback from a small user group without including objective performance metrics of the AI model during deployment.

Besides, the paper could benefit from some adjustments to improve the ease of reading, especially for a broader academic audience or those less familiar with all the technical details.

---

### Official Review · Reviewer_z8CZ · 2024-10-03
**Deployment of Deep Learning Model in Real World Clinical Setting: A Case Study in Obstetric Ultrasound**

**Confidence:** 1

**Summary:**

The paper introduces a framework for the real-world deployment of deep learning models in medical workflows. The aim is to facilitate real-world clinical validation of AI in medical imaging in clinical settings with integration challenges and complex medical workflows. The framework is designed to allow early-stage testing of AI models and to provide real-time feedback to users. Key components include: (i) HDMI-to-USB converter for capturing real-time ultrasound video feeds, (ii) docker containers to isolate and manage different aspects of the system, and (iii) a wireless display of AI predictions to clinicians via tablets. The framework is evaluated on one particular application: fetal ultrasound standard plane detection in obstetrics. This evaluation was conducted in a hospital where inexperienced medical students and experienced practitioners used the system during live ultrasound sessions.

**Strengths:**

The paper demonstrates an understanding of the practical challenges of AI deployment in clinical settings. It carefully documents the process of creating the proposed deployment framework. The methodology section is technically sound providing detailed justifications for key design choice. While most AI studies tend to focus on performance in lab environments, this paper emphasizes the practical implementation in a clinical setting which is both valuable and potentially novel for the particular application (a deep learning model for fetal ultrasound). The focus on containerization and isolating the research code from clinical infrastructure is a smart approach that addresses deployment robustness in clinical settings. The authors also ensure reproducibility by offering access to a public code repository. Finally: The paper is well-written, clearly organized, and organized in a manner that supports understanding and implementation.

**Weaknesses:**

The results section lacks comparative analysis with established clinical methods or other AI deployment approaches. There’s limited inclusion and discussion of performance metrics which should be important for evaluating the proposed deployment framework. Instead, the feedback is mostly qualitative, focusing on user experiences rather than objective scientific data. The authors should provide a comparison between their deployment framework and similar frameworks in other medical imaging applications. This could either be in terms of technical performance (e.g., lower latency) or usability (e.g., easier to implement in hospitals).

There’s nothing particularly new in the method itself: no novel algorithms, architectures, or breakthroughs in model performance. The paper applies existing techniques (e.g., the PCBM model for ultrasound image detection), and the claimed novelty comes from the framework implementation, rather than scientific discovery. The framework’s components are mostly engineering solutions rather than novel scientific contributions. A more thorough comparative analysis and discussion could possibly make this paper more than just an engineering report.

**Final Rebuttal Confidence:**

2

**Final Rebuttal Justification:**

I remain unconvinced about the paper's scientific contribution. If the primary contribution is the proposed framework, it requires a more thorough and systematic evaluation. If the focus is on a case study, the paper should emphasize this aspect and provide a deeper qualitative analysis. However, I reiterate that case studies are outside my area of expertise. After reviewing their rebuttal, I am inclined to uphold my recommendation for rejection.

**Justification:**

The paper is sufficiently strong on framework motivation and documentation but lacking in validation and scientific novelty. The paper presents a well-implemented solution to a deployment problem, but it reads more like an engineering report rather than a contribution of new knowledge to the field of AI in medical imaging.

---

### Official Review · Reviewer_BEct · 2024-10-09
**Framework for running AI models for ultrasound in clinical environment**

**Confidence:** 4

**Summary:**

The paper presents a computational framework for running and assessing AI models for ultrasound images, lowering the barrier for deployment of AI in clinical use. The authors use obstetric ultrasound as a case study to demonstrate the framework. The core of the work is description of the design and implementation of this framework, aiming to make it generic. The actual AI model implemented using the framework for the case study is standard plane detector for fetal ultrasound images. Although framed as presenting a generic framework, the study is more a description of a case study.

**Strengths:**

Development of AI models in biomedical applications is booming, but there is an obvious gap in implementation of such modes to real world clinical environment. Thus, a framework lowering the barrier and facilitating such integration to enable rapid testing and feedback is a warmly welcome idea. The attempt to make the framework generic is positive, although more could be done to actually show it is such.

**Weaknesses:**

Weight is placed on generic design, but only one case study is presented. It remains unclear whether any other modalities or even application within the domain of ultrasound imaging could be easily tucked in using this framework. It would be great to show other examples, at least give specs on limitations (what type of other modalities, limits to image sizes etc.), or tone down the framing as a generic platform.

Further, the case study is merely descriptive, for example, the authors state they wanted to study whether the explainability of the Ai model in the case studied provided added value, but this answer to this limits to anecdotes by the users. No systematic way to collect the feedback or quantitative values using the framework is presented, leaving the idea that the implementation falls a bit short of the aims.

**Justification:**

Although not fully convincing as a generic framework for implementing AI systems in clinical environments, the study nevertheless takes a valuable step towards lowering the barrier or bridging the gap in implementing AI models for actual use, which is a much welcomed and valuable effort.

---

### Meta-Review · Area_Chair_zKW7 · 2024-11-03

**Recommendation:** Reject
**Confidence:** 4

**Metareview:**

The submitted paper proposes a framework for deploying AI models in clinical routine. The presented framework was used for a case study in which a AI-based standard plane detection model for fetal ultrasound was deployed and tested by clinicians. All reviewers acknowledged that the translation of AI models into clinical routine is indeed an important aspect and early deployment is beneficial. The objective and the workflow of the method is described clearly and to gather clinical feedback is regarded highly relevant.
However, the reviewers raised questions on whether the presented work has enough scientific contribution and novelty. Among others, reviewers noted the lack of details when the framework (i.e. the converter) and the model were described as well as the lack of standardization when evaluating user feedback. The quantitative measurements were focusing on the latency when using the presented pipeline in comparison to the direct execution. Both runtimes seem to be slow considering the application. The qualitative feedback from the clinicians seems to only consider the model output and not the proposed pipeline or the way it is deployed. Some of the reviewers felt that the presented paper is rather as a case study and since the proposed solution is agnostic of the deployed (AI) method, out of scope for NLDL. The reviewers suggested to add more comparisons with other deployment solutions, more than one deployed models, and a standardized feedback questionnaire for clinical evaluation. Taken into consideration all strengths and weaknesses, the submitted paper can unfortunately not be suggested for acceptance for NLDL. The authors are advised to follow the feedback of the reviewers, extend their manuscript and may select a better targeted conference for publication.

**Suggested Changes To The Recommendation:**

2: I'm certain of the recommendation.  It should not be changed

---

### Decision · Program_Chairs · 2024-11-06

Reject